# Prevalence and associated factors with long COVID in the Brazilian population: The role of health-related behaviors and sociodemographic characteristics

José Francisco Martoreli Júnior[1]*, Andrey O. Pedroso[1], Laís do E. S. Lima[1], Cristine Maria P. Gusmão[2], Mayra G. Menegueti[1], Hemílio F. C. Coêlho[3], Cristina Mara Zamarioli[1], Ana Cristina de O. e Silva[4], Glenda R. O. Naiff[5], Elucir Gir[1], Renata K. Reis[1]

1 General and Specialized Nursing Department, University of São Paulo at Ribeirão Preto College of Nursing, Ribeirão Preto, São Paulo, Brazil, 2 Amélia Uchoa Campus, Tiradentes University Center, Maceió, Alagoas, Brasil, 3 Center for Exact and Natural Sciences, Federal University of Paraíba, João Pessoa, Paraíba, Brasil, 4 Health Sciences Center, Federal University of Paraíba, João Pessoa, Paraíba, Brasil, 5 Institute of Health Science,Federal University of Pará, Belém, Pará, Brasil

☯ These authors contributed equally to this work.
* jose.martoreli@usp.br

**Data availability statement:** All relevant data are within the manuscript and its Supporting Information files.

## Abstract

The disease caused by the 2019 coronavirus (COVID-19) has resulted in unprecedented morbidity and mortality worldwide, with many individuals experiencing persistent symptoms and a decline in quality of life after infection. This study aims to analyze the prevalence and associated factors of long COVID in the Brazilian population, focusing on disease severity and immunization status. This observational, cross-sectional web survey employed a quantitative approach to analyze data from 4,231 participants, focusing on the prevalence and associated factors of long COVID. Data were analyzed using inferential statistical methods to identify factors associated with the outcome. A multivariable logistic regression model was applied to determine the variables independently associated with long COVID. To determine the best model, a stepwise selection method was employed, and the model's performance was assessed utilizing a Receiver Operating Characteristic Curve (ROC). The findings revealed a long COVID prevalence of 56.4% (2,386 cases), with men having a 36.46% (OR = 1,36 CI = 1,17−1,58) higher chance of developing long COVID compared to women. A prior diagnosis before vaccination increased by 22.30% (OR = 1,22 CI = 1,05−1,41). Additionally, the use of sedatives and alcohol was linked to increases of 24.50% (OR = 1,24 CI = 1,07−1,43) and 34.95% (OR = 1,34 CI = 1,02−1,75), respectively. Beneficiaries of social programs faced a 47.29% (OR = 1,47 CI = 1,27−1,70) higher, while individuals with comorbidities had a 33.47% (OR =1,33 CI = 1,20−1,48). Hospitalization significantly raised the likelihood of prolonged symptoms by 331.92% (OR = 4,31 CI = 2,53−7,87). Overall, various factors, including

**Funding:** This study was financed in part by the Coordenação de Aperfeiçoamento de Pessoal de Nível Superior - Brasil (CAPES) - Finance Code 001 and CAPES - EPIDEMICS, Emergency Selection Notice IV CAPES, Impacts of the Pandemic. The funders had no role in study design, data collection and analysis, decision to publish, or preparation of the manuscript.

**Competing interests:** The authors have declared that no competing interests exist.

sedative and alcohol use, were factors associated with long COVID, whereas vaccination showed a positive impact, suggesting that association models can help healthcare professionals identify high-risk patients and tailor care effectively.

## Introduction

The disease caused by the 2019 coronavirus (COVID-19) has resulted in unprecedented morbidity and mortality worldwide. In Brazil, as of October 2024, the number of individuals affected by COVID-19 reached 38,953,513, with 713,626 deaths and over 38 million recoveries [1]. Additionally, data from the Ministry of Health indicate that the number of cases and deaths has impacted different regions of the country variably, with the Central-West region showing the highest incidence of cases at 28,037.8 per 100,000 inhabitants and the highest death rate at 412.4 per 100,000 inhabitants as of October 2024, revealing the significant socio-spatial inequalities that characterize the country [2].

Regarding the impact of COVID-19 among recovered individuals, scientific and clinical evidence is evolving concerning the subacute and long-term effects of this disease, which can affect multiple systems and organs. Reports suggest effects of COVID-19 infection, with the main symptoms being fatigue, muscle pain, cough, shortness of breath, chest pain, cognitive disturbances, difficulties in concentration, dizziness, headache, anxiety, depression, arthralgia, and a decline in quality of life [3–8].

Cellular damage, a robust innate immune response with the production of inflammatory cytokines, and a pro-coagulant state induced by COVID-19 infection may contribute to these sequelae. Survivors of previous coronavirus infections, including the SARS epidemic of 2003 and the Middle East Respiratory Syndrome (MERS) outbreak of 2012, exhibited persistent symptoms, reinforcing concerns about clinically significant sequelae from COVID-19. Based on recent literature, long COVID is divided into two categories: (1) subacute or ongoing symptomatic COVID-19, which includes symptoms and abnormalities present from four to 12 weeks beyond acute COVID-19; and (2) chronic syndrome or post-COVID-19, which includes persistent symptoms and abnormalities present beyond 12 weeks of the onset of acute COVID-19 and not attributable to alternative diagnoses [4,5,9].

Many individuals experience persistent symptoms and a decline in quality of life after COVID-19 infection. Hospitalized individuals reported symptoms up to 110 days post-onset in the UK [3].

With no proven treatments or rehabilitation guidance, long COVID affects people's ability to return to normal life and work. The societal impact, in terms of increased health burdens and economic and productivity losses, is substantial. Long COVID presents a pressing public health challenge [3].

Clearly, this condition is a public health concern. There are varying definitions of long COVID, making it difficult to estimate its true prevalence worldwide. According to the World Health Organization (WHO), it is believed that at least 17 million

people in the European Region experienced long COVID within the first two years of the pandemic. Data from the Centers for Disease Control and Prevention (CDC) regarding the U.S. population indicate that the prevalence of long COVID among non-institutionalized adults aged 18 and older decreased from 7.5% in June 2022 to 6.0% in June 2023, and from 18.9% to 11.0% among adults reporting a prior COVID-19 infection. After an initial decline, the prevalence has remained stable since 2023, with approximately one-quarter of adults with long COVID reporting significant activity limitations [10].

Despite increasing global attention to long COVID, evidence from Latin American countries remains limited. Brazil presents unique epidemiological, social, and healthcare characteristics that may influence long COVID outcomes, highlighting the need for context-specific data. Recent studies conducted in Brazil and across Latin America demonstrate substantial prevalence and long-term impact of the condition [11–13]

Therefore, long COVID has been increasingly recognized as a global public health concern, national evidence in Brazil remains limited. Brazil has a unique epidemiological profile, social inequalities, vaccination strategies and healthcare access dynamics. Recent research conducted in Brazil has begun to shed light on the prevalence and long-term impacts of the condition and regional data from broader Latin American populations highlight the potential implications for healthcare demand and patient quality of life [11,12].

By generating population-based evidence, this study contributes to filling this knowledge gap and provides relevant support for healthcare planning and the development of strategies aimed at early identification and management of individuals at risk of long COVID making the objective of this investigation to evaluate the prevalence and associated factors of long COVID in the Brazilian population.

## Materials and methods

### Ethical aspects

This study was conducted with the approval of the Ethics Committee of Hospital Eduardo de Menezes (Protocol CAAE: 65929522.1.0000.5183). All participants or their legal representatives provided digital informed consent to be included in this study.

Regarding confidentiality and privacy, it is emphasized that these is maintained. The information is confidential, and participants is not identified at any time. All data is stored for a minimum period of five years and is disposed of in accordance with Brazilian Law No. 12,965/2014.

### Study design and setting

Web survey study, observational, cross-sectional, and analytical with a quantitative approach [14].

The adapted questionnaire of the "COVID-19 Global Clinical Platform: Case Report Form for Post-COVID-19 Condition," developed by the World Health Organization (WHO), was used in this study [15]. This instrument was designed to systematically gather standardized clinical data from individuals who had experienced COVID-19, with the goal of contributing to the WHO Clinical Platform and supporting global initiatives for understanding, monitoring, and managing post-COVID-19 condition. It includes questions covering sociodemographic characteristics, pre-existing comorbidities, the clinical course of the acute infection, and subsequent symptoms persisting beyond the acute phase [15].

However, to meet the specific objectives of this study and to better capture the psychosocial and functional aspects of long COVID in a Brazilian population, additional elements were incorporated into the adapted questionnaire [16–17].

Notably, the adapted tool extends beyond the original scope by incorporating validated scales specifically, the Female Sexual Quotient (QS-F) and nd the EQ-5D Brazilian population norms to assess questions exploring subjective experiences such as mental health symptoms (e.g., sadness, anxiety, concentration difficulties), sleep quality, appetite and weight changes, substance use, and limitations in activities of daily living. These dimensions, absent from the original

WHO form, reflect an intentional effort to evaluate the broader psychosocial and functional impacts of post-COVID condition [16–17].

The full questionnaire can be provided upon request to the authors.

## Sample and data collection procedures

The reference population includes all Brazilians residing in Brazil at the time of the survey who had access to the questionnaire disseminated through social media (Facebook, Twitter, Instagram, WhatsApp, or email). Additionally, only participants aged 18 years or older who were diagnosed with COVID-19 between 2020 and 2023 were included.

The data collection was between 01/05/2023 and 31/08/2023. For participant recruitment, the Respondent-Driven Sampling (RDS) method adapted for online environments was used [14]. Initially, data collection was conducted by selected researchers from all regions of Brazil. In total 39 of these researchers underwent four hours of pre-training to conduct an online survey. The RDS method used in this study was implemented as follows: a random selection of a set of participants (seeds) was made. The seeds were limited to ten referrals each in the first selection, and they managed this data in an Excel spreadsheet. The referral slots were limited to ensure enough candidates remained in the pool to continue the referral chain as much as possible within their networks. After the seeds made their referrals, each participant who returned contact via WhatsApp was interviewed and received a similar training to manage the spreadsheet with the ten referrals to be made. A new round of selection was generated when the referral spreadsheets were returned to the seeds. This back-and-forth replaced physical coupons and allowed data collection to be managed remotely.

Additionally, invitations to participate in the survey were disseminated through social media (Facebook, Twitter, Instagram, WhatsApp, or email). The data collection questionnaire was converted to a digital format. Invitations for participation in the survey were sent through messages on the aforementioned social media platforms, and the survey link was also posted. The link containing the questionnaire was made available on the researchers' social media profiles. It is emphasized that social media was used only as a means to publicize the research, ensuring that the collected information was solely that provided by the participant.

For the operationalization of data collection, participants received a link that included information about the nature and confidentiality of the research. By clicking on the link, participants were directed to the REDCap platform, where they accessed a question about participation in the study. Upon acceptance, participants were given access to the Informed Consent Form (ICF) and the study questionnaire.

The ICF for online data collection was available on the homepage, and participants could only access the questionnaire if they agreed to participate in the research by selecting the option "I have read and agree to participate in this research," thereby giving their informed consent. In case of non-acceptance, the participant was directed to a closure page with thanks for their attention.

It is noteworthy that participants were guaranteed the right to have a second copy of the ICF, as it was made available for download, allowing them to complete it at their convenience. It is further highlighted that each Internet Protocol (IP) address could submit only one response.

## Data source

The questionnaire was administered online using a structured electronic survey developed through Google Forms. The survey link was disseminated via widely used digital platforms, including WhatsApp, Instagram, and Facebook, enabling broad reach and accessibility.

A pilot test was conducted where 19 individuals were invited to complete a preliminary version of the questionnaire. Feedback was received, where participants provided suggestions and comments about the survey content and format.

Adjustments were made to improve clarity, relevance, and cultural appropriateness, and the finalized version was then used for data collection in the main study.

The minimal anonymized dataset is published in the supporting information for replicability.

## Study variables

The outcome variable in this study was the presence of symptoms lasting for four weeks or more ("Did you have symptoms for four weeks or more?") following the initial onset of COVID-19 symptoms. This binary variable (yes/no) was used to classify individuals as meeting an operational definition of Long COVID (also referred to as Post-COVID Condition) based on the prevailing clinical and epidemiological criteria at the time of data collection. Participants were asked: "Did you have symptoms for four weeks or more?" This threshold (≥4 weeks) was commonly employed by public health authorities such as the U.S. Centers for Disease Control and Prevention (CDC) and national surveillance systems during the early phases of the pandemic, prior to the formal case definition established by the World Health Organization (WHO) in October 2021. At that time, this four-week criterion served as a practical marker for persistent symptoms and was widely adopted in observational studies investigating the prolonged effects of COVID-19 [18,19].

This approach aligns with WHO recommendations that emphasize excluding alternative explanations for ongoing symptoms. Although our study used a pragmatic threshold of four weeks due to its relevance at the time, we recognize that more recent guidance, including the WHO's Delphi-based consensus, defines Post-COVID Condition as symptoms occurring three months from onset and lasting at least two months [18,19].

The independent variables were social and demographic characteristics which selected variables were: Education (No education/Never completed any grade, Incomplete elementary education, Complete elementary education, Incomplete high school, Complete high school, Incomplete higher education, Complete higher education, Specialization, Masters, Doctorate, Postdoctoral), Sex (Female, Male), Confirmed diagnosis of COVID-19 (1 time, 2 times, 3 times, 4 times, 5 or more), Diagnosed with COVID-19 before vaccination (No, Yes), Chronic illness diagnosis before COVID-19 (No, Yes, I don't know), Received antibiotics (No, Yes), Received prescribed antivirals (Yes, No), Received Ivermectin (No, Yes), Received chloroquine (No, Yes), Received home medicine (No, Yes), Received sedatives (No, Yes), Beneficiary of social programs (No, Yes), Income (Less than 1 minimum wage, 1 minimum wage, 2 minimum wages, 3 minimum wages, 4 minimum wages), Depression or Anxiety (Yes, No), Alcoholic drinks (No, Yes), Occupation (Unemployed, Employed, Self-Employed, Retired, Student).

## Statistical analysis

The statistical analysis followed a structured approach beginning with descriptive statistics to summarize the characteristics of the study population. Frequencies and percentages were calculated for all categorical variables. Subsequently, Fisher's exact test was applied to assess associations between the outcome variable and the other qualitative variables in the dataset. For variables with at least one cell in the columns or rows containing a value of zero, the test could not be performed, and therefore they were not included in the model.

A logistic regression model was then constructed to estimate which of these variables demonstrated evidence of an association with the outcome of interest. To obtain evidence of the best model, the stepwise model selection method was considered, which automatically selects factors associated with an outcome within the context of a regression model. This method combines features of two other variable selection methods: forward selection, generally referred to as the forward method, and backward selection, commonly known as the backward method. The model's performance was assessed utilizing a Receiver Operating Characteristic Curve (ROC).

The curve shows the relationship between sensitivity (true positive) and specificity (1 – false positive) at different classification thresholds with an AUC of 0.7.

In addition to evaluating the discriminative ability of the model using the ROC curve and AUC, the calibration of the logistic regression model was assessed using the Hosmer–Lemeshow goodness-of-fit test. This test verifies whether the predicted probabilities of the outcome adequately reflect the observed frequencies across deciles of risk. A non-significant p-value

indicates that the model is well calibrated, whereas a significant result suggests poor fit. Although the ROC curve assesses the model's ability to discriminate between individuals with and without the outcome, the Hosmer–Lemeshow test answers a different question, namely whether the estimated probabilities are accurate when compared to the real data distribution.

All analyses were performed using the statistical software R and Jamovi obtained for free [20,21].

## Results

In total, 5,950 people were interviewed. However, with the aim of researching only those who responded to the symptom item over a period of four weeks or more in the questionnaire, a filter was applied to the database, resulting in a final sample of 4,231 people who effectively answered the question related to this variable. Table 1 shows the distribution of these responses.

Fisher's exact test was subsequently performed using the dichotomous variable "Have you had symptoms for a period of 4 weeks or more?" in comparison with the other variables in the form, resulting in p-values < 0.05 for the variables listed in Table 2.

Note that some participants did not respond to all questions presented in this table; consequently, the total number of responses may vary for each variable.

Participants who reported symptoms lasting ≥4 weeks differed significantly from those without persistent symptoms in terms of sex, previous health status, medication received, and infection history. Although females predominated in both groups, those with long-lasting symptoms had a markedly higher proportion of males (23.3% vs 10.4%; p < 0.01). Individuals reporting prolonged symptoms were also more likely to have had confirmed COVID-19 reinfection (≥2 episodes: 39.5% vs 27.91%; p < 0.01) and to have been diagnosed prior to vaccination (63.6% vs 53.3%; p < 0.01). Moreover, the presence of pre-existing chronic illnesses was almost twice as frequent in this group (14.5% vs 8.4%; p < 0.01).

Participants with persistent symptoms more frequently received antibiotics (65.2% vs 47.4%; p < 0.01) and Ivermectin (40.35% vs 33.04%; p < 0.01), as well as chloroquine (11.15% vs 5.4%; p < 0.01). Socioeconomic differences were modest but statistically significant, with a higher proportion earning one minimum wage among those with persistent symptoms (39.55% vs 35.05%; p < 0.01). Mental health was also notably affected, with almost half of individuals with prolonged symptoms reporting depression or anxiety (44.4% vs 25.0%; p = 0.017). Additionally, alcohol consumption was slightly more frequent in this group (62.96% vs 57.73%; p = 0.031).

The results indicate, at a 95% confidence level, that the following variables are associated with the outcome variable (p-value < 0.05): occupation, social program beneficiary status, income, number of COVID-19 diagnoses, COVID-19 diagnosis prior to vaccination, presence of chronic illness, depression or anxiety, receipt of antibiotics, receipt of prescribed antivirals, receipt of ivermectin, receipt of chloroquine, receipt of home remedies, and consumption of alcoholic beverages and sedatives.

A logistic regression model was then generated to estimate which of these variables provide evidence of being associated with the outcome of interest. Considering the association of these variables with the outcome variable.

Table 3 presents the results obtained from fitting the logistic regression model to the data, using the stepwise selection method. For each variable included in the final model, the specific response category associated with the outcome.

**Table 1. Distribution of interviewees according to indication of presence of symptoms for a period of four weeks or more.**

| Have you had symptoms for a period of 4 weeks or more? | Frequency | Relative frequency (Prevalence estimate in %) | Lower limit CI (%) | Upper limit CI (%) | Margin of error, in percentage, plus or minus |
|---|---|---|---|---|---|
| No | 1845 | 43,6 | 42,1 | 45,1 | 1,49 |
| Yes | 2386 | 56,4 | 54,9 | 57,9 | 1,49 |
| Total | 4231 | 100 | – | – | |

**Table 2. Results of association tests between the outcome variable and the other variables considered for the study.**

| Variables | | Have you had symptoms for a period of 4 weeks or more? | | | | p-value |
|---|---|---|---|---|---|---|
| | | **No** | | **Yes** | | |
| | | *n* | % | *n* | % | |
| Education | No education/Never completed any grade | 2 | 0.11% | 6 | 0.25% | < 0,001 |
| | Incomplete elementary education | 21 | 1.15% | 43 | 1.81% | |
| | Complete elementary education | 23 | 1.25% | 36 | 1.52% | |
| | Incomplete high school | 63 | 3.44% | 73 | 3.07% | |
| | Complete high school | 410 | 22.37% | 540 | 22.69% | |
| | Incomplete higher education | 551 | 30.06% | 732 | 30.77% | |
| | Complete higher education | 372 | 20.30% | 484 | 20.36% | |
| | Specialization | 260 | 14.19% | 309 | 12.99% | |
| | Masters | 95 | 5.18% | 124 | 5.21% | |
| | Doctorate | 29 | 1.58% | 28 | 1.18% | |
| | Postdoctoral | 7 | 0.38% | 5 | 0.21% | |
| Sex | Female | 1,224 | 89.61% | 1,125 | 76.67% | < 0,001 |
| | Male | 142 | 10.39% | 343 | 23.33% | |
| How many times have you had a confirmed diagnosis of COVID-19? | 1 time | 1,326 | 72.11% | 1,441 | 60.47% | < 0,001 |
| | 2 times | 417 | 22.68% | 692 | 29.03% | |
| | 3 times | 77 | 4.19% | 198 | 8.31% | |
| | 4 times | 13 | 0.71% | 37 | 1.55% | |
| | 5 or more | 6 | 0.33% | 15 | 0.63% | |
| You were diagnosed with COVID-19 before being vaccinated? | No | 830 | 46.74% | 839 | 36.44% | < 0,001 |
| | Yes | 947 | 53.26% | 1,463 | 63.56% | |
| Before having COVID-19, you were diagnosed with a chronic illness? | No | 1,604 | 87.96% | 1,907 | 80.51% | < 0,001 |
| | Yes | 153 | 8.39% | 343 | 14.48% | |
| | I don't know | 67 | 3.67% | 119 | 5.02% | |
| Did you receive antibiotics? | No | 909 | 52.64% | 818 | 34.30% | < 0,001 |
| | Yes | 928 | 47.36% | 1,556 | 65.20% | |
| Received prescribed antivirals? | Yes | 52 | 9.26% | 85 | 8.64% | < 0,001 |
| | No | 509 | 90.74% | 924 | 91.36% | |
| You received Ivermectina? | No | 1,219 | 66.96% | 1,423 | 59.65% | < 0,001 |
| | Yes | 613 | 33.04% | 943 | 40.35% | |
| You received chloroquine? | No | 1,675 | 94.56% | 2,093 | 87.73% | < 0,001 |
| | Yes | 146 | 5.44% | 266 | 11.15% | |
| You received home medicine? | No | 1,279 | 69.57% | 1,428 | 59.86% | < 0,001 |
| | Yes | 561 | 30.43% | 944 | 39.63% | |
| Did you receive sedatives? | No | 1,692 | 89.78% | 2,092 | 87.67% | < 0,001 |
| | Yes | 98 | 10.22% | 209 | 12.33% | |
| Beneficiary of Social Programs? | No | 1,696 | 92.04% | 2,102 | 88.12% | < 0,001 |
| | Yes | 128 | 7.96% | 245 | 11.88% | |
| Income | Less than 1 minimum wage | 300 | 16.27% | 407 | 17.06% | < 0,001 |
| | 1 minimum wages | 646 | 35.05% | 944 | 39.55% | |
| | 2 minimum wages | 332 | 18.01% | 384 | 16.08% | |
| | 3 minimum wages | 88 | 4.78% | 89 | 3.73% | |
| | 4 minimum wages | 479 | 25.99% | 562 | 23.58% | |

*(Continued)*

**Table 2.** (Continued)

| Variables | | Have you had symptoms for a period of 4 weeks or more? | | | | p-value |
|---|---|---|---|---|---|---|
| | | **No** | | **Yes** | | |
| | | *n* | % | *n* | % | |
| Depression or Anxiety | Yes | 32 | 75.00% | 112 | 44.44% | 0.017 |
| | No | 39 | 25.00% | 69 | 17.07% | |
| Alcoholic Drinks | No | 748 | 41.27% | 884 | 37.04% | 0.031 |
| | Yes | 1,047 | 57.73% | 1,422 | 62.96% | |
| What is your occupation? | 1 – Unemployed | 139 | 10.10% | 231 | 9.68% | 0.027 |
| | 2 – Employed | 866 | 62.61% | 1,060 | 44.45% | |
| | 3 – Self- Employed | 229 | 16.55% | 286 | 11.99% | |
| | 4 – Retired | 65 | 4.70% | 62 | 2.60% | |
| | 5 – Student | 505 | 36.46% | 701 | 29.37% | |

**Table 3.** Result of adjusting the logistic regression model and calculating prevalence ratios.

| Variable | Ref. | Estimate | p-value | Odds Ratio | C.I. for Odds Ratio (95%) | | Prevalence Ratio | C.I. for P.R. (95%) | |
|---|---|---|---|---|---|---|---|---|---|
| | | | | | Inferior Limits | Superior Limits | | Inferior Limits | Superior Limits |
| Intercept | – | −1,37440 | < 0,001 | – | – | – | – | – | – |
| Sex Male | Female | 0,31085 | < 0,001 | 1,36 | 1,17 | 1,58 | 1,13 | 1,06 | 1,20 |
| You were diagnosed with COVID-19 before being vaccinated? Yes | No | 0,20131 | < 0,001 | 1,22 | 1,05 | 1,41 | 1,08 | 1,02 | 1,14 |
| Did you receive sedatives? Yes | No | 0,21913 | < 0,001 | 1,24 | 1,07 | 1,43 | 1,09 | 1,02 | 1,15 |
| Alcoholic Drinks Yes | No | 0,2997 | 0,0321 | 1,34 | 1,02 | 1,75 | 1,11 | 1,01 | 1,22 |
| Beneficiary of Social Programs Yes | No | 0,38725 | < 0,001 | 1,47 | 1,27 | 1,70 | 1,15 | 1,09 | 1,21 |
| Before having COVID-19, you were diagnosed with a chronic illness? Yes | No | 0,28873 | < 0,001 | 1,33 | 1,20 | 1,48 | 1,13 | 1,07 | 1,20 |
| Did you receive antibiotics? Yes | No | 0,33275 | < 0,001 | 1,39 | 1,20 | 1,61 | 1,14 | 1,07 | 1,21 |
| Did you receive prescribed antivirals? Yes | No | −0,26623 | < 0,001 | 1,30 | 1,12 | 1,51 | 1,10 | 1,04 | 1,17 |
| Have you heard of the term Long COVID Yes | No | 0,43400 | < 0,001 | 1,54 | 1,31 | 1,81 | 1,17 | 1,10 | 1,24 |
| Have you sought any health service to investigate and/or treat these symptoms? Yes | No | 0,51449 | < 0,001 | 1,67 | 1,44 | 1,93 | 1,22 | 1,15 | 1,29 |
| Have you ever been hospitalized to treat these symptoms? Yes | No | 1,46308 | < 0,001 | 4,31 | 2,53 | 7,87 | 1,48 | 1,35 | 1,63 |
| Employed | Unemployed | −0,26946 | 0,0445 | 0,76 | 0,51 | 0,99 | 0,89 | 0,81 | 0,99 |
| Self- Employed | Unemployed | −0,35574 | < 0,001 | 0,70 | 0,541 | 1,00 | 0,88 | 0,77 | 1,00 |
| Retired | Unemployed | −0,74578 | < 0,001 | 0,47 | 0,29 | 0,77 | 0,70 | 0,54 | 0,91 |
| Student | Unemployed | −0,17636 | 0,2011 | 0,83 | 0,63 | 1,09 | 0,93 | 0,83 | 1,03 |
| You received home medicine? Yes | No | 0,26532 | < 0,001 | 1,30 | 1,12 | 1,51 | 1,10 | 1,04 | 1,17 |

At the 95% confidence level, the multivariate logistic regression model demonstrated that male participants had a 36% higher likelihood of reporting symptoms lasting ≥4 weeks when compared to females (adjusted OR = 1.36; 95% CI: 1.17–1.58). Individuals diagnosed with COVID-19 before vaccination showed a 22% increased likelihood of long-duration

symptoms (OR = 1.22; 95% CI: 1.05–1.41). Reporting use of sedatives was also associated with a 24% higher chance of long COVID (OR = 1.24; 95% CI: 1.07–1.43), as well as consuming alcoholic beverages (OR = 1.34; 95% CI: 1.02–1.75). Beneficiaries of social programs had a 47% increased likelihood of prolonged symptoms (OR = 1.47; 95% CI: 1.27–1.70).

Participants who self-reported a pre-existing chronic illness had a 33% greater probability of long COVID (OR = 1.33; 95% CI: 1.20–1.48). Those who received antibiotics also showed a 39% increase in risk (OR = 1.39; 95% CI: 1.20–1.61). Conversely, individuals with a history of prescribed antiviral use demonstrated a reduction in the chance of persistent symptoms (OR = 0.77; inverse of 1.30, 95% CI: 0.66–0.90), suggesting a possible protective effect.

Awareness of the term "long COVID" was associated with a 54% higher probability of reporting prolonged symptoms (OR = 1.54; 95% CI: 1.31–1.81), and seeking healthcare services for these symptoms increased this likelihood by 67% (OR = 1.67; 95% CI: 1.44–1.93). Hospitalization for symptoms presented the strongest association, with participants being over four times more likely to report long COVID (OR = 4.31; 95% CI: 2.53–7.87).

Regarding occupational status, using "unemployed" individuals as the reference category, being employed was associated with a 24% reduction in the likelihood of long COVID (OR = 0.76; 95% CI: 0.51–0.99), and being retired showed a 53% lower probability (OR = 0.47; 95% CI: 0.29–0.77). "Self-employed" individuals had a borderline reduction in risk (OR = 0.70; 95% CI: 0.54–1.00), whereas the "student" category did not show a statistically significant association. Finally, reporting the use of home medicine during illness was linked to a 30% increased likelihood of persistent symptoms (OR = 1.30; 95% CI: 1.12–1.51).

Once the interpretations have been made, Fig 1 presents the Receiver Operating Characteristic Curve (ROC) of the adjusted model.

## Discussion

This study provides a comprehensive overview of the factors associated with persistence of COVID 19 symptoms. Key findings indicate that demographic, clinical, and behavioral characteristics influence the likelihood of developing long COVID.

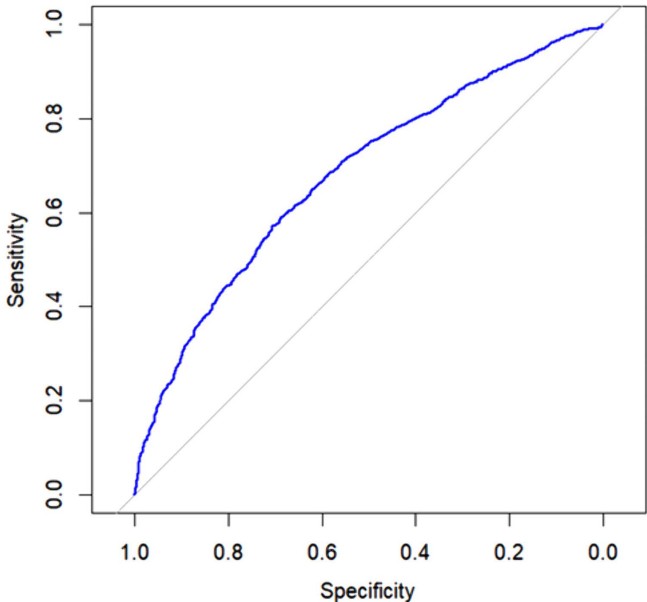

**Fig 1. ROC Curve of the Adjusted Logistic Regression Model.** (X axis) show the Specificity of the model; (Y axis) show the Sensitivity of the model.

The persistence of COVID-19 (Long COVID) is widely recognized as a continuing global public health challenge, with prevalence and associated factor studies extensively published across various countries, including Brazil. Our work contributes to this crucial context by providing a detailed and timely analysis of associated factors within a Brazilian population. This specific contribution is vital for informing public health policies and underscoring the importance of continuous monitoring of the post-COVID-19 condition.

It was observed that men have a 36.46% higher chance of developing what is termed long COVID compared to women (OR = 1.36 CI = 1.17–1.58). With a prevalence ratio of 1.13 (95% CI: 1.06–1.20) in line with the literature that evidences gender differences in immune responses to COVID-19 infection [22,23]. Several biological mechanisms may underlie this disparity males are known to mount more robust innate inflammatory responses, which can lead to increased tissue damage during acute infection, whereas women tend to exhibit prolonged adaptive immune activation, potentially offering more effective viral clearance and long-term protection [22,23]. However, these findings must be interpreted with caution due to the unequal gender distribution in the sample, which may have introduced bias and limited the robustness of sex-related comparisons.

Another relevant finding was the 22.30% increase in the likelihood of prolonged symptoms in individuals who had COVID-19 before being vaccinated (OR = 1.22 CI = 1.05–1.41) and a prevalence ratio of 1.08 (95% CI: 1.02–1.14). Vaccination has proven effective in reducing the severity and duration of the disease, and contracting the infection before being immunized may result in a less effective immune response, favoring persistent symptoms [24–26]. Consistent with data showing two pre-infection mRNA doses reduce long COVID risk [27–28].

Additionally, both the use of sedatives and alcohol consumption were identified as associated factors. Those who used sedatives had a 24.50% increase in the probability of suffering from prolonged symptoms (OR = 1.24 CI = 1.07–1.43), while those consuming alcohol showed a 34.95% higher chance of having long COVID (OR = 1.34 CI = 1.02–1.75). The use of sedatives and alcohol consumption, with prevalence ratios of 1.09 (95% CI: 1.02–1.15) and 1.11 (95% CI: 1.01–1.22), respectively, suggest that psychoactive substances like alcohol may impair the immune response and prolong recovery [29].

The study also pointed out that beneficiaries of social programs are 47.29% more susceptible to long COVID (OR = 1.47 CI = 1.27–1.70) with a prevalence ratio of 1.15 (95% CI: 1.09–1.21). This finding may be related to barriers to accessing adequate healthcare that affect low-income populations, compromising timely and effective treatment of the disease [30].

The presence of comorbidities, such as chronic diseases, raised the chances of prolonged symptoms by 33.47% (OR = 1.33 CI = 1.20–1.48) with a prevalence ratio of 1.13 (95% CI: 1.07–1.20). Patients with diseases such as diabetes and hypertension are already considered more vulnerable to COVID-19, which may explain their slower recovery [31].

The use of antibiotics (OR = 1.39 CI = 1.20–1.61) and antivirals (OR = 1.30 CI = 1.12–1.51) was also associated with a higher association of persistent symptoms. These medications may be indicators of the severity of the initial infection, but their use may be linked to subsequent complications, such as secondary infections or adverse effects [32–36].

Regarding occupation, employed individuals (OR = 0.76 CI = 0.58–0.99) and retirees (OR = 0.47 CI = 0.29–0.77) showed a lower probability of prolonged symptoms compared to the unemployed, which may reflect greater financial stability and access to healthcare. Additionally, the use of home remedies was associated with a 30.38% increase of prolonged symptoms (OR = 1.30 CI = 1.12–1.51), suggesting that self-medication without medical guidance may be ineffective and potentially delay appropriate treatment. Some studies have discussed the use of home remedies for the treatment and prevention of COVID-19, while also cautioning that most of these practices lack scientific validation [35,36].

Individuals who were hospitalized for COVID-19 presented a 331.92% increase in the probability of persistent symptoms (OR = 4.31 CI = 2.53–7.87), which is consistent with findings from studies showing that, six months after symptom onset, the most frequently reported complaints among recovered patients were fatigue or muscle weakness and sleep disturbances, these individuals who experienced more severe forms of the disease exhibited a higher prevalence of psychological complications, such as anxiety and depression, as well as impaired pulmonary diffusion capacity [37,38]. These

outcomes highlight the long-term health burden of COVID-19 and emphasize the need for structured post-discharge care, particularly for patients with severe illness. Continued follow-up in larger cohorts is essential to fully characterize the range and duration of post-COVID health consequences

The analysis of the ROC curve of the adjusted model, with an AUC of 0.7, demonstrates good predictive capacity, while the Hosmer-Lemeshow test ($p < 0.05$) shows that the model is suitable for estimating the likelihood of symptom occurrence for four weeks or more. Such findings reinforce the need for public health policies that address socioeconomic inequalities and include preventive strategies targeting vulnerable groups.

The limitations of this study include selection bias due to reliance on a web-based survey that may underrepresent certain demographics, recall bias from self-reported data, and response bias in reporting behaviors like alcohol or sedative use. Additionally, the cross-sectional design limits causal inference, and potential confounding variables, such as mental health status, were not measured. These factors may affect the direction and magnitude of the observed associations, suggesting that the findings should be interpreted cautiously. Addressing these limitations in future research could improve the validity of conclusions regarding long COVID.

Several factors associated with an increased association of long COVID have been identified, such as the use of sedatives, alcohol consumption, and comorbidities, as well as the positive role of vaccination against COVID-19. Furthermore, using association models like those presented in this study may help healthcare professionals identify patients at higher risk and adapt care more effectively.

The results of this study provide a detailed analysis of the interrelations between demographic, clinical, and behavioral factors with the persistence of COVID-19 symptoms for more than four weeks, highlighting the inherent complexity of the condition known as long COVID. Crucial factors for the exacerbation and prolongation of symptoms include the use of antibiotics and antivirals, prior hospitalization, unvaccinated individuals, consumers of alcohol and sedatives, as well as those with comorbidities and beneficiaries of social programs.

Conversely, it was observed that individuals in more stable occupational situations, as well as those who resort to traditional therapies, showed a lower probability of persistent symptoms, suggesting a direct relationship between socioeconomic factors and the clinical outcome of long COVID. The predictive robustness of the adjusted model, corroborated by the ROC curve analysis, underscores the need for more targeted preventive and therapeutic strategies, especially for vulnerable population groups.

## Conclusion

This study offers valuable insights into the prevalence and associated factors of long COVID among the Brazilian population, revealing that male sex, prior COVID-19 infection before vaccination, and the use of sedatives and alcohol are associated with prolonged symptoms. However, caution is warranted due to limitations such as potential selection and response biases from the web-based survey, reliance on self-reported data, and the cross-sectional design, which limits causal interpretations. Despite these challenges, the findings show the importance of socioeconomic factors and emphasizes a need for targeted preventive and therapeutic strategies, particularly for vulnerable populations, and underscores the importance of an integrated, multidimensional approach to healthcare that helps professionals identify high-risk patients and adapt care effectively highlighting areas for future research to enhance understanding and healthcare strategies regarding long COVID.

## Supporting information

**S1 File. STROBE.**
(DOCX)

**S2 File. Minimal anonymized dataset.**
(XLSX)

## Acknowledgments

The authors would like to thank the Foundation Coordination for the Improvement of Higher Education Personnel and all participants of the study.

## Author contributions

**Conceptualization:** Laís do E. S. Lima, Cristine Maria P. Gusmão, Hemílio F. C. Coêlho, Cristina Mara Zamarioli, Ana Cristina de O. e Silva, Glenda R. O. Naiff, Elucir Gir, Renata K. Reis.

**Data curation:** José Francisco Martoreli Júnior, Andrey O. Pedroso, Laís do E. S. Lima, Cristine Maria P. Gusmão, Hemílio F. C. Coêlho, Ana Cristina de O. e Silva, Glenda R. O. Naiff, Renata K. Reis.

**Formal analysis:** José Francisco Martoreli Júnior, Hemílio F. C. Coêlho.

**Funding acquisition:** José Francisco Martoreli Júnior, Cristine Maria P. Gusmão, Ana Cristina de O. e Silva, Glenda R. O. Naiff, Elucir Gir, Renata K. Reis.

**Investigation:** José Francisco Martoreli Júnior, Andrey O. Pedroso, Laís do E. S. Lima, Cristine Maria P. Gusmão, Mayra G. Menegueti, Hemílio F. C. Coêlho, Ana Cristina de O. e Silva, Glenda R. O. Naiff, Elucir Gir, Renata K. Reis.

**Methodology:** José Francisco Martoreli Júnior, Mayra G. Menegueti, Hemílio F. C. Coêlho, Ana Cristina de O. e Silva, Glenda R. O. Naiff, Elucir Gir, Renata K. Reis.

**Project administration:** José Francisco Martoreli Júnior, Cristine Maria P. Gusmão, Hemílio F. C. Coêlho, Ana Cristina de O. e Silva, Glenda R. O. Naiff, Elucir Gir, Renata K. Reis.

**Resources:** José Francisco Martoreli Júnior, Andrey O. Pedroso, Laís do E. S. Lima, Cristine Maria P. Gusmão, Cristina Mara Zamarioli, Ana Cristina de O. e Silva, Glenda R. O. Naiff, Renata K. Reis.

**Software:** José Francisco Martoreli Júnior, Andrey O. Pedroso, Hemílio F. C. Coêlho, Renata K. Reis.

**Supervision:** José Francisco Martoreli Júnior, Andrey O. Pedroso, Ana Cristina de O. e Silva, Glenda R. O. Naiff, Elucir Gir, Renata K. Reis.

**Validation:** José Francisco Martoreli Júnior, Andrey O. Pedroso, Cristine Maria P. Gusmão, Cristina Mara Zamarioli, Ana Cristina de O. e Silva, Glenda R. O. Naiff, Elucir Gir, Renata K. Reis.

**Visualization:** José Francisco Martoreli Júnior, Andrey O. Pedroso, Mayra G. Menegueti, Hemílio F. C. Coêlho, Cristina Mara Zamarioli, Ana Cristina de O. e Silva, Glenda R. O. Naiff, Elucir Gir, Renata K. Reis.

**Writing – original draft:** José Francisco Martoreli Júnior, Mayra G. Menegueti, Hemílio F. C. Coêlho, Cristina Mara Zamarioli, Ana Cristina de O. e Silva, Glenda R. O. Naiff, Renata K. Reis.

**Writing – review & editing:** José Francisco Martoreli Júnior, Andrey O. Pedroso, Laís do E. S. Lima, Cristine Maria P. Gusmão, Mayra G. Menegueti, Cristina Mara Zamarioli, Ana Cristina de O. e Silva, Glenda R. O. Naiff, Elucir Gir, Renata K. Reis.

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
