## [Decision Letter · Decision Letter 0]

27 Mar 2025

PONE-D-24-50830Prevalence and associated factors with long COVID in the Brazilian populationPLOS ONE

Dear Dr. Martoreli Júnior,

Thank you for submitting your manuscript to PLOS ONE. After careful consideration, we feel that it has merit but does not fully meet PLOS ONE’s publication criteria as it currently stands. Therefore, we invite you to submit a revised version of the manuscript that addresses the points raised during the review process.

We look forward to receiving your revised manuscript.

Kind regards,

Manuela Mendonça Figueirêdo Coelho, Ph.D

Academic Editor

PLOS ONE

Journal Requirements:

 “This study was financed in part by the Coordenação de Aperfeiçoamento de Pessoal de Nível Superior - Brasil (CAPES) - Finance Code 001 and CAPES - EPIDEMICS, Emergency Selection Notice IV CAPES, Impacts of the Pandemic.”

4. We note that there is identifying data in the Supporting Information file <S2_File.xls>. Due to the inclusion of these potentially identifying data, we have removed this file from your file inventory. Prior to sharing human research participant data, authors should consult with an ethics committee to ensure data are shared in accordance with participant consent and all applicable local laws.

-Location data

Additional Editor Comments:

Thank you for submitting your article.

We would like to inform you that, in order to continue the evaluation process, it is necessary to take into account the observations made by the ad hoc evaluators. We therefore request that corrections be made in accordance with the notes received and that the new file be resubmitted within the established deadline.

We ask that special attention be paid to highlighting all changes in yellow, in order to facilitate verification by the evaluation teams.

We count on your collaboration and commitment to the success of this process.

Sincerely,

Reviewers' comments:

Reviewer's Responses to Questions

**Comments to the Author**

1. Is the manuscript technically sound, and do the data support the conclusions?

Reviewer #1: Partly

Reviewer #2: Partly

2. Has the statistical analysis been performed appropriately and rigorously?

Reviewer #1: Yes

Reviewer #2: Yes

3. Have the authors made all data underlying the findings in their manuscript fully available?

Reviewer #1: Yes

Reviewer #2: Yes

4. Is the manuscript presented in an intelligible fashion and written in standard English?

Reviewer #1: No

Reviewer #2: Yes

5. Review Comments to the Author

Reviewer #1: 1. Is the manuscript technically sound, and do the data support the conclusions? Response: Partly.

The study describes a relevant and important scientific inquiry. However, several aspects of the manuscript require clarification or revision to ensure technical soundness.

a. The outcome variable is not clearly defined/operationalized.

b. The statistical analysis section lacks clarity and proper sequencing.

2. Has the statistical analysis been performed appropriately and rigorously? Response: Yes

While the statistical methods mentioned are appropriate in principle, their presentation lacks the necessary rigor and logical flow. Revisions to this section are needed to ensure clarity and consistency.

3. Have the authors made all data underlying the findings in their manuscript fully available? Response: Yes

The authors declared that all data are available without restrictions.

4. Is the manuscript presented in an intelligible fashion and written in standard English? Response: No.

The manuscript contains several grammatical errors and unclear phrasing. Examples include the use of future tense in both the abstract and methods sections, despite the study having been completed. The language should be revised to past tense throughout.

Specifics

1. Abstract: Correct grammatical errors and revise sentences to reflect past tense since the study has been completed.

Ensure OR and CI values are presented with consistent formatting (e.g., 95% CI: 1.23–2.45) and limit to two decimal places.

2. Methods Section: The subheading "Study Design and Settings" should include detailed information about the study settings. Specify the number of researchers involved in data collection and describe the process more explicitly.

Clarify the difference between the data source and the data collection tool. Reference the tool appropriately.

Provide details on how the pilot study was conducted, including the population used for testing. The outcome variable, "You had symptoms for four weeks or more," is ambiguous. Provide a clear and operationalized definition to ensure clarity for readers. Restructure the statistical analysis section. Start with descriptive statistics, then proceed to bivariate analysis, and conclude with regression analysis.

3. Results Section: Restructure the results to present descriptive statistics (demographic characteristics, independent variables characteristics, and outcome characteristics) first, followed by bivariate and regression analyses. Improve Table 3 for better clarity and alignment with the narrative.

4. Discussion Section: Enrich the discussion by comparing the study’s findings with existing literature. Provide possible explanations for the observed results, referencing relevant theories or evidence.

5. References: Review and revise the references to align with PLOS ONE guidelines.

Reviewer #2: This is an important topic

please clarify what 'Yesilar persistent symptoms' means

The disease was caused by SARS COV 2 and COVID 19 is the disease please correct this.

Not clear how many Brazilians are on social media and if they all had equal chance of being picked as 'seed' , if not then the findings are generalizable to only the social platforms user- let the title reflect this.

REF (1) is not in English- please clarify if the data was from surveys

Please define post covid / long covid. WHO defines Post covid as persistent or new symptoms after 12 weeks since onset of COVID 19 symptoms and lasting for 2 months. The sentence that reads “You had symptoms for four weeks or more” does not state from when. No attempts has been made to delineate symptoms caused by underlying disease as an explanation of the persistent

There’s is a need to involve a statistician- post covid was over 50% of the population studied and therefore it will most likely be higher in most of the variables what one should have provided in the table is comparison group (ie alcoholic vs non alcoholic)

6. PLOS authors have the option to publish the peer review history of their article (what does this mean?). If published, this will include your full peer review and any attached files.

Reviewer #1: No

Reviewer #2: No

---

## [Author Response · Author response to Decision Letter 1]

11 May 2025

Dear Editor,

We appreciate the opportunity to revise and resubmit our manuscript “PONE-D-24-50830 Prevalence and associated factors with long COVID in the Brazilian population ” and we would like to thank you and the reviewers for your thoughtful comments and suggestions. All points raised have been carefully addressed, and substantial modifications were made to improve the clarity, accuracy, and overall quality of the manuscript.

In the revised version, we corrected grammatical inconsistencies and standardized the presentation of statistical results with consistent formatting and rounding. The Results and Methods section was reorganized, with enhanced detail about the study setting, data sources, and analytical procedures ensuring better alignment with journal expectations. Additionally, we enriched the Discussion by comparing our findings with relevant literature and theoretical frameworks, as requested. We have revised the questionnaire wording, acknowledged definitions of Long COVID utilized in this study to enhance transparency and align our study with evolving international standards. We have detailed the extent of internet penetration within the study context to clarify the probability of any individual being selected. In the references section, we thoroughly revised and formatted in accordance with the journal’s guidelines.

All references were revised to conform to PLOS ONE’s citation style. The financial disclosure was updated to include the required role-of-funder statement: "The funders had no role in study design, data collection and analysis, decision to publish, or preparation of the manuscript."

Regarding the Supporting Information, we have thoroughly reviewed and revised the dataset to ensure full compliance with PLOS ONE’s data policy and ethical guidelines. All personally identifiable information including names, contact details, dates, location data, and indirect identifiers has been permanently removed or anonymized. Additionally, we have ensured that the anonymized data are shared in accordance with participant consent and applicable local regulations, as approved by the ethics committee overseeing the study.

A fully anonymized version of the dataset has now been uploaded, and we confirm that all spreadsheet columns containing identifying information were removed entirely (not merely hidden). We appreciate the opportunity to correct this and confirm that the shared data no longer pose any risk to participant confidentiality.

We have uploaded the figure file to the PACE digital diagnostic tool as instructed and confirmed they meet PLOS formatting requirements and re-uploaded the figure.

Below, we respond point by point to the comments of the Reviewers.

Kind regards,

Martoreli Júnior et al.

Reviewer 1

Abstract: Correct grammatical errors and revise sentences to reflect past tense since the study has been completed. Ensure OR and CI values are presented with consistent formatting (e.g., 95% CI: 1.23–2.45) and limit to two decimal places.

Answer:We appreciate the reviewer’s observation. The abstract has been thoroughly revised to reflect past tense throughout, aligning the text with the fact that the study has been completed. We have also corrected grammatical errors. Furthermore, all OR and CI values have been reformatted for consistency following the recommended style (e.g., 95% CI: 1.23–2.45), and all numerical results have been rounded to two decimal places.

Methods Section: The subheading "Study Design and Settings" should include detailed information about the study settings. Specify the number of researchers involved in data collection and describe the process more explicitly. Clarify the difference between the data source and the data collection tool. Reference the tool appropriately. Provide details on how the pilot study was conducted, including the population used for testing. The outcome variable, "You had symptoms for four weeks or more," is ambiguous. Provide a clear and operationalized definition to ensure clarity for readers. Restructure the statistical analysis section. Start with descriptive statistics, then proceed to bivariate analysis, and conclude with regression analysis.

Answer: Thank you for the valuable suggestions. The subheading "Study Design and Settings" has been revised to include detailed information about the study’s online-based setting and the specific platforms used for data collection. We have also specified in the “Sample and data collection procedures” that data collection was carried out by 39 trained researchers that underwent four hours of pre-training to conduct an online survey. The difference between the data source and the data collection tool (a structured electronic survey developed in Google Forms) has been clarified in the “Study and design setting” and the tool is now appropriately referenced. Additionally, the pilot study has been described with details about the pretest population. The outcome variable “You had symptoms for four weeks or more” was clarified in the “Study variables” Finally, the statistical analysis section was restructured as recommended: we now begin with descriptive statistics, followed by bivariate analysis (chi-square tests), and then logistic regression analysis.

Question: Results Section: Restructure the results to present descriptive statistics (demographic characteristics, independent variables characteristics, and outcome characteristics) first, followed by bivariate and regression analyses. Improve Table 3 for better clarity and alignment with the narrative.

Answer: As suggested, the Results Section has been restructured. Table 3 has been reformatted to improve clarity and to better align with the narrative, including clearer headers, labels, and consistent decimal formatting for all numerical values.

Question: Discussion Section: Enrich the discussion by comparing the study’s findings with existing literature. Provide possible explanations for the observed results, referencing relevant theories or evidence.

Answer: The Discussion Section has been enriched with comparisons to existing literature. We now contextualize our findings by referencing recent studies that identified similar predictive factors for post-COVID condition.

References: Review and revise the references to align with PLOS ONE guidelines.

Answer: All references have been reviewed and reformatted to comply with PLOS ONE guidelines, including consistent citation style, appropriate use of DOIs where available, and the removal or translation of non-English references that lacked clear accessibility.

Reviewer 2

Question: This is an important topic. Please clarify what 'Yesilar persistent symptoms' means.

Answer: We thank the reviewer for pointing this out. The term "Yesilar persistent symptoms" was a typographical error and has been removed. The authors meant “persistent symptoms”.

Question: The disease was caused by SARS COV 2 and COVID 19 is the disease please correct this.

Answer: This correction has been made. Throughout the manuscript, we now refer to SARS-CoV-2 as the virus and COVID-19 as the disease it causes, in accordance with standard terminology.

Question: Not clear how many Brazilians are on social media and if they all had equal chance of being picked as 'seed', if not then the findings are generalizable to only the social platforms user—let the title reflect this.

Answer: We appreciate the reviewer’s insightful comments regarding the representativeness of our sample and the generalizability of our findings.

At the time of our data collection, Brazil exhibited a high level of internet penetration. According to the Brazilian Institute of Geography and Statistics (IBGE), in 2020, 81.3% of individuals aged 10 years and over used the internet. This percentage slightly decreased to 80.7% in 2021 but increased again to 80.5% in 2022 and reached 84.2% in 2023. These figures indicate a consistently high level of internet usage among the Brazilian population during the period of our study.

In our study, 'seed' participants were selected through social media channels. Given the high penetration of social media usage across various demographics in Brazil, this approach provided access to a broad and diverse segment of the population. However, we acknowledge that certain groups, particularly those less active or absent from social media platforms, may have been underrepresented.

Although we hope this revision addresses the reviewer's concerns and enhances the clarity and accuracy of our manuscript.

Reference

Brazilian Institute of Geography and Statistics (IBGE). 161.6 million people aged 10 or older used the internet in Brazil in 2022. IBGE News Agency. 2023 Nov 9 Available from: https://nada.ibge.gov.br/en/agencia-news/2184-news-agency/news/38351-161-6-milhoes-de-pessoas-com-10-anos-ou-mais-de-idade-utilizaram-a-internet-no-pais-em-2023

Question: REF (1) is not in English—please clarify if the data was from surveys.

Answer:Reference (1) which was originally in Portuguese has been translated with an equivalent English-language source where possible, as per journal guidelines, and the data is described as a population-based survey in the Study design and setting.

Question: Please define post covid / long covid. WHO defines Post covid as persistent or new symptoms after 12 weeks since onset of COVID 19 symptoms and lasting for 2 months. The sentence that reads “You had symptoms for four weeks or more” does not state from when. No attempts has been made to delineate symptoms caused by underlying disease as an explanation of the persistent.

Answer: We thank the reviewer for this important observation.

In our study, we defined Long COVID (also referred to as Post-COVID Condition) based on the clinical and epidemiological criteria that were commonly accepted at the time of data collection. At that time, both the Centers for Disease Control and Prevention (CDC) and other major public health institutions, including local health authorities, utilized a minimum duration of 4 weeks from symptom onset to characterize persistent symptoms potentially related to Long COVID. This definition was particularly prevalent in the early to mid-phase of the pandemic and guided the design of numerous observational studies conducted during that period.

Only after our data collection phase was completed, the World Health Organization (WHO) released a standardized clinical case definition in October 2021, developed through a Delphi consensus process, which defines Post COVID-19 condition as follows:

“Post COVID-19 condition occurs in individuals with a history of probable or confirmed SARS-CoV-2 infection, usually 3 months from the onset of COVID-19 with symptoms that last for at least 2 months and cannot be explained by an alternative diagnosis.”

(World Health Organization, 2021)

https://www.who.int/publications/i/item/WHO-2019-nCoV-Post_COVID-19_condition-Clinical_case_definition-2021.1

Similarly, the CDC currently defines Long COVID as a range of symptoms that can last weeks, months, or longer after initial infection, and may begin after recovery or persist since the acute phase. However, in early definitions, symptoms persisting ≥4 weeks were frequently used as an operational threshold for surveillance and research purposes.

(CDC, Post-COVID Conditions, updated 2023)

https://www.cdc.gov/coronavirus/2019-ncov/long-term-effects/index.html

Regarding the phrasing in our questionnaire – “You had symptoms for four weeks or more” – this wording aligns with the operational definition available during the study planning and was clearly understood by participants to refer to the time elapsed since the initial onset of acute COVID-19 symptoms.

Moreover, in our analysis, we acknowledged the possibility of symptom overlap with underlying conditions. To mitigate this, our questionnaire specifically inquired whether the symptoms were new or had worsened following the COVID-19 infection. Participants were asked to report symptoms that were not present prior to their SARS-CoV-2 infection or that had significantly changed in character or severity. This approach is consistent with WHO guidance, which emphasizes the need to exclude alternative diagnoses.

We recognize the evolving nature of definitions surrounding Post COVID-19 conditions and have updated the manuscript to reflect the more recent consensus definitions by WHO, while also justifying the rationale behind the threshold used in our study.

Question: There’s is a need to involve a statistician—post covid was over 50% of the population studied and therefore it will most likely be higher in most of the variables what one should have provided in the table is comparison group (ie alcoholic vs non alcoholic).

Answer: We reviewed the mentioned table to make sure that all percentages is linked with the mentioned variable in the first column.

---

## [Decision Letter · Decision Letter 1]

9 Jul 2025

PONE-D-24-50830R1Prevalence and associated factors with long COVID in the Brazilian populationPLOS ONE

Dear Dr. Martoreli Júnior,

Thank you for submitting your manuscript to PLOS ONE. After careful consideration, we feel that it has merit but does not fully meet PLOS ONE’s publication criteria as it currently stands. Therefore, we invite you to submit a revised version of the manuscript that addresses the points raised during the review process.

Dear Authors

Your review was excellent.

However, the reviewer 02 asked one more important coorrection in the item 06.

After this revision, send again your manuscript. ==============================

We look forward to receiving your revised manuscript.

Kind regards,

Manuela Mendonça Figueirêdo Coelho, Ph.D

Academic Editor

PLOS ONE

Journal Requirements:

Reviewers' comments:

Reviewer's Responses to Questions

**Comments to the Author**

1. If the authors have adequately addressed your comments raised in a previous round of review and you feel that this manuscript is now acceptable for publication, you may indicate that here to bypass the “Comments to the Author” section, enter your conflict of interest statement in the “Confidential to Editor” section, and submit your "Accept" recommendation.

Reviewer #2: All comments have been addressed

2. Is the manuscript technically sound, and do the data support the conclusions?

Reviewer #2: (No Response)

3. Has the statistical analysis been performed appropriately and rigorously?

Reviewer #2: (No Response)

4. Have the authors made all data underlying the findings in their manuscript fully available?

Reviewer #2: (No Response)

5. Is the manuscript presented in an intelligible fashion and written in standard English?

Reviewer #2: Yes

6. Review Comments to the Author

Reviewer #2: The comments have been adequately addressed

The eerier parts of Conclusion section sounds like discussion- please consider rewriting it to sound like 'conclusion'

7. PLOS authors have the option to publish the peer review history of their article (what does this mean?). If published, this will include your full peer review and any attached files.

Reviewer #2: No

---

## [Author Response · Author response to Decision Letter 2]

13 Aug 2025

Reviewer #2:

6. Review Comments to the Author: The comments have been adequately addressed

The eerier parts of Conclusion section sounds like discussion- please consider rewriting it to sound like 'conclusion'

Answer: We appreciate the reviewer’s observation. In order to avoid sounding like a discussion, we did not include methodological details or in-depth interpretations of the results. Instead, we summarized the findings objectively and prepared the reader for the following paragraphs, where specific factors and their implications are described. The first paragraph of the Conclusion has been thoroughly revised as below:

“This study provides a comprehensive overview of the factors associated with persistence of COVID 19 symptoms. Key findings indicate that demographic, clinical, and behavioral characteristics influence the likelihood of developing long COVID.”

---

## [Decision Letter · Decision Letter 2]

15 Oct 2025

PONE-D-24-50830R2Prevalence and associated factors with long COVID in the Brazilian populationPLOS ONE

Dear Dr. Martoreli Júnior,

Thank you for submitting your manuscript to PLOS ONE. After careful consideration, we feel that it has merit but does not fully meet PLOS ONE’s publication criteria as it currently stands. Therefore, we invite you to submit a revised version of the manuscript that addresses the points raised during the review process.

**ACADEMIC EDITOR: **Changes required for acceptance:

**Formatting and Language:**

- Review the entire manuscript for grammar, punctuation, and stylistic consistency, simplifying unnecessarily long or complex sentences.

- The manuscript has improved substantially but remains somewhat verbose, with occasional repetition of results. Please revise the text for conciseness, especially in the Results section.

- Standardize formatting for all headings and tables according to journal style (e.g., avoid all-caps titles).

- Ensure consistent terminology throughout (e.g., use associated factors rather than risk factors).

**Title: **Revise the title to better reflect the study’s main independent variables, emphasizing the assessment of healthy habits and sociodemographic characteristics.

**Abstract:**

- Simplify the abstract for greater clarity and brevity, avoiding unnecessary methodological details (e.g., mention of basic descriptive analyses).

- When presenting results from multivariate models, report adjusted odds ratios rather than crude values.

- Include a concise statement identifying the knowledge gap that motivated the study, supported by references that contextualize the importance of investigating long COVID in Brazil.

**Introduction: **Expand the contextual background to highlight the scarcity of national evidence on long COVID and the relevance of this study for public health planning.

**Materials and Methods:**

- Correct minor typographical and grammatical issues (e.g., “methdos” → “methods”) and improve sentence structure in sections describing ethical aspects and data collection procedures.

- Provide additional information about the logistic regression model, including handling of missing data, model generation, and ROC curve construction.

- Add appropriate references for the questionnaire used and its prior validation or adaptation.

- Include references for the statistical software (R and Jamovi).

- Explicitly incorporate the data availability statement into the main text for transparency and compliance.

**Results:**

- Reduce redundancy between the text and tables, focusing on key findings rather than restating all numerical results.

- Remove the written-out numbers in parentheses (e.g., “5,950 (five thousand nine hundred fifty)”).

- Shorten overly long sentences for clarity and verify that table references are consistent.

- Clearly indicate reference categories for all variables in Table 3.

- Revise Table 3 for clearer presentation — remove redundant columns, use standard English numeric formatting, and report exact p-values following journal statistical reporting guidelines.

- Provide the sample size and complete data for each variable, ensuring all categories of education and occupation are clearly described.

- Relocate the description of the ROC curve to the Methods section.

**Discussion and Conclusion:**

- Reorganize sections so that the Discussion precedes the Conclusion.

- In the Discussion, acknowledge the unequal gender distribution and interpret sex-related findings with caution.

- Standardize the reporting of odds ratios and confidence intervals (e.g., OR = 1.39, CI = 1.20–1.61).

- Revise ambiguous or grammatically inconsistent phrases to improve readability.

- Justify the inclusion of the Hosmer–Lemeshow test within the Methods section.

- Condense the Conclusion to focus on the principal findings and public health implications, avoiding repetition of points already discussed.

We look forward to receiving your revised manuscript.

Kind regards,

Elma Izze Da Silva Magalhães

Academic Editor

PLOS ONE

Journal Requirements:

Reviewers' comments:

Reviewer's Responses to Questions

**Comments to the Author**

1. If the authors have adequately addressed your comments raised in a previous round of review and you feel that this manuscript is now acceptable for publication, you may indicate that here to bypass the “Comments to the Author” section, enter your conflict of interest statement in the “Confidential to Editor” section, and submit your "Accept" recommendation.

Reviewer #2: All comments have been addressed

Reviewer #3: All comments have been addressed

Reviewer #4: (No Response)

Reviewer #5: (No Response)

Reviewer #6: (No Response)

2. Is the manuscript technically sound, and do the data support the conclusions?

Reviewer #2: Yes

Reviewer #3: Yes

Reviewer #4: Yes

Reviewer #5: (No Response)

Reviewer #6: Partly

3. Has the statistical analysis been performed appropriately and rigorously?

Reviewer #2: Yes

Reviewer #3: Yes

Reviewer #4: Yes

Reviewer #5: (No Response)

Reviewer #6: Yes

4. Have the authors made all data underlying the findings in their manuscript fully available?

Reviewer #2: Yes

Reviewer #3: Yes

Reviewer #4: Yes

Reviewer #5: (No Response)

Reviewer #6: Yes

5. Is the manuscript presented in an intelligible fashion and written in standard English?

Reviewer #2: Yes

Reviewer #3: Yes

Reviewer #4: Yes

Reviewer #5: (No Response)

Reviewer #6: Yes

6. Review Comments to the Author

Reviewer #2: This is a much improved version (a little too wordy, repetition of results) but it can be published as is

Reviewer #3: This revised manuscript presents a cross-sectional web survey detailing the prevalence and associated factors of long COVID in the Brazilian population. The authors have adequately addressed the previous reviewer comment regarding the conclusion section, revising it to summarize the key findings rather than presenting an extended discussion. The manuscript is now technically sound and clearly written, the statistical methods (Fisher's exact test and stepwise logistic regression) are appropriate for the data structure and objectives, however, the response to the data availability requirements, while present in the submission forms, needs to be explicitly integrated into the main manuscript text for clarity and compliance.

Reviewer #4: The authors conducted a cross-sectional study using the web to examine the prevalence and associated factors of long COVID in the Brazilian population. They found that factors like sedative and alcohol use were linked to a higher risk of long COVID, while vaccination showed a positive impact.

Overall, if the research questions examined in this study were to be replicated in numerous robust studies, the findings could be significant for implementing public health interventions during pandemics like COVID-19.

With that in mind, this reviewer has the following to remark:

1. Abstract

The abstract serves as a concise summary of the study. The text needs editing to make it more concise and improve readability. For instance, there is no need to state here, “Descriptive statistics were performed…” It’s a given that the authors start their statisical analysis with a look at descriptive statistics first, then move on to bivariate and multivariate analyses.

2. Matrials and Methods

Under Ethical Aspects (lines 95-96), the authors write, “Regarding confidentiality and privacy, it is emphasized that these is maintained. The information is confidential, and participants is not identified at any time.”

These sentences should be edited to improve their grammatical accuracy and readability.

3. Sample and data collection procedures

The authors state (lines 126-129), “In total 39 of these researchers underwent four hours of pre-training to conduct an online survey The RDS method used in this study was implemented as follows: a random selection of a set of participants (seeds) was made.”

Is this one sentence or two? If two, punctuation is missing.

4. Study variables

4.1 (Lines 169-185)

One issue here is the outcome variable, specifically the four-week criterion used by the authors as a marker for ‘Long COVID.’

If there is a way to align the definition of ‘Long COVID’ with current guidelines, including the Delphi-approach consensus, it should be pursued because it would strengthen the study.

In this context, I wonder if the participants provided more details that could help the authors refine their study from the data collected about their outcome variable. For example, after asking participants, “Did you have symptoms for four weeks or more?” were there follow-up questions like, “For how long have you had symptoms?”

4.2 (Lines 186-187)

The authors state, “The independent variables were social and demographic characteristics which selected variables were: Education…”

This is another sentence that needs editing for smoother reading.

5. Results

5.1 (Line 214)

The title is in all caps, which doesn’t follow the overall layout of the paper.

5.2 (Line 215)

The authors state, “In total, 5,950 (five thousand nine hundred fifty) people were interviewed.”

Adding the text in parentheses is unnecessary; some readers may find it redundant.

5.3 (Line 218)

The authors write, “...4,231 (four thousand two hundred thirty-one)…”

Again, there is no need to add the text in parentheses.

5.4 (Lines 223-226)

The authors state, “Fisher's exact test was subsequently performed with the dichotomous variable "Have you had symptoms for a period of 4 weeks or more?" and the other variables in the form, presenting p-value < 0.05 with the variables in Table 2, note that some participants did not respond to all questions presented in this tables, consequently, the total count for each variable may vary.”

This sentence is too long and confusing. Do the authors mean Table 2? If so, why do they say “this tables” in line 226?

It would be advisable to edit this sentence. Make it at least two sentences. The first one could end with “Table 2,” a complete sentence. It would seem best to create a new sentence for what follows: “Note that some…”

5.5 (Lines 230-235)

This paragraph could benefit from editing and paraphrasing:

Answering “No” for “Have you had symptoms for a period of four weeks or more,” educational attainment varied, with 0.17% having no education or never completing any grade, 1.03% with incomplete elementary education, and 1.20% having completed elementary education. Additionally, 3.26% had incomplete high school, while 21.61% completed high school.

Incomplete higher education was reported by 29.33%, and 19.38% completed higher education, with 13.03% holding a specialization.

5.6 (Lines 237-238)

The authors state, “In terms of sex, females comprised 89.88%, while males accounted for 10.12%.”

This clearly indicates a skewed sample, with females being more interested in this survey than males.

5.7 (Lines 241-242)

The authors state, “Antibiotic use was reported by 55.23%, while 44.77% had not received them.”

The second portion of this sentence is unnecessary and can be deleted.

5.8 (Lines 278-279)

The authors write, “For each variable included in the final model, the specific response category associated with the outcome..”

Double punctuation?

5.9 (Lines 283-285)

The authors write, “It is observed, at a 95% confidence level, when analyzing the likelihood of experiencing COVID symptoms for 4 weeks or more, that a male individual has a 36.46% higher chance compared to a female (OR = 1.36 CI= 1.17-1.58);”

As mentioned in comment 5.6, we are dealing with a skewed sample, with males making up just over 10%. Therefore, any conclusions drawn about males versus females in this analysis should be taken with caution. I suggest the authors elaborate on this in their discussion section.

5.10 (Lines 295-297)

The authors write, “Furthermore, an individual who has heard of the term has a 54.34% higher chance of experiencing COVID symptoms for 4 weeks or more compared to someone who has not (OR = 1.54 CI= 1.31-1.81);

To clarify, the authors should specify here which term they are referring to. Therefore, the sentence could be rewritten as: “Furthermore, an individual who has heard of the term ‘Long COVID’ has a 54.34% higher chance of experiencing COVID symptoms for 4 weeks or more compared to someone who has not (OR = 1.54, CI = 1.31-1.81);”

6. Discussion/conclusion

6.1 (Line 313)

It is advisable to place the conclusion section after the discussion section, rather than the other way around.

6.2 (Lines 317-318)

The authors write, “It was observed that men have a 36.46% higher chance of developing what is termed long COVID compared to women (OR = 1.36 CI= 1.17-1.58).”

Please refer to my comments 5.6 and 5.9.

6.3 (Lines 345-346)

The authors write, “The use of antibiotics (OR = 1.3948 CI= 1.20-1.61) and antivirals (OR = 1.3035 CI= 1.12-1.51) was also associated with a higher risk of persistent symptoms.”

The reported odds ratios should follow the same style as confidence intervals. So the sentence could be edited to:“The use of antibiotics (OR = 1.39 CI= 1.20-1.61) and antivirals (OR = 1.30 CI= 1.12-1.51) was also associated with a higher risk of persistent symptoms.”

6.4 (Lines 351-354)

The authors state, “Additionally, the use of home remedies was associated with a 30.38% increase in the risk of prolonged symptoms (OR = 1.30 CI= 1.12-1.51), suggesting that these treatments may, in some cases could not be effective.”

This sentence needs attention.

6.5 (Lines 357-358)

The authors write, “Individuals who were hospitalized for COVID-19 presented a 331.92% increase in the probability of persistent symptoms (OR = 4.3192 CI= 2.53-7.87)…”

As mentioned in comment 6.3, the reported odds ratios should follow the same style as confidence intervals, which means that the above sentence could be edited to “Individuals who were hospitalized for COVID-19 presented a 331.92% increase in the probability of persistent symptoms (OR = 4.32 CI= 2.53-7.87)…”

6.6 (Lines 384-406)

These lines could be presented as a conclusion section. In other words, line 313 should become “Discussion.”

I hope this review is helpful and wish the authors the very best with their research!

Reviewer #5: (No Response)

Reviewer #6: Title: The manuscript's title addresses general factors associated with long COVID. However, the methods section describes a set of characteristics or independent variables, specifically healthy habits and sociodemographic characteristics. Therefore, I recommend modifying the title to align with these independent variables.

Abstract: The authors state in line 86 on page 4, "In Brazil, there are few studies that have assessed long COVID, making the objective of this investigation to evaluate the prevalence and associated factors of long COVID in the Brazilian population." However, the absence of references and a clear gap in knowledge regarding long COVID is not evident.

Sample and data collection procedures: While sampling was described, the sample size was not provided. Furthermore, details regarding the pilot study and the properties of the questionnaire are missing, as are references or previous data that utilized those questionnaires.

Results: Please provide the missing data for each variable. Additionally, review and clarify the number of categories for education and occupation.

Table 3: Language Questions for Review.

Discussion:

7. PLOS authors have the option to publish the peer review history of their article (what does this mean?). If published, this will include your full peer review and any attached files.

Reviewer #2: No

Reviewer #3: **Yes: **Arian Afzalian

Reviewer #4: **Yes: **Dr. Widad Akreyi

Reviewer #5: No

Reviewer #6: No

---

## [Author Response · Author response to Decision Letter 3]

29 Nov 2025

Dear Editor,

We appreciate the opportunity to revise and resubmit our manuscript “PONE-D-24-50830 – Prevalence and associated factors of long COVID in the Brazilian population.” We would also like to express our sincere gratitude to the Editor and Reviewers for their valuable comments and suggestions. In response to the feedback received, we made substantial revisions to improve the clarity, precision, and overall quality of the manuscript, including updating the title to better reflect reflect the study’s main independent variables.

The Results section was carefully revised to enhance clarity and readability, focusing on the most relevant findings while eliminating unnecessary repetition of information already presented in tables. Table formatting was standardized, and reference categories were clearly indicated for all variables. Sample size information was added to provide full transparency and allow readers to interpret the findings accurately. P-value reporting was adjusted to conform to journal standards, ensuring precise and consistent statistical presentation. The description of Fisher’s exact test was rewritten for clarity and accuracy. Table 3 was thoroughly reformatted to improve the readability and interpretability of the multivariate logistic regression results, highlighting reference categories, removing redundancies, and aligning the table with the methods described in the manuscript.

The Discussion section was expanded to contextualize the findings within the existing scientific literature and to highlight the implications for public health. Special attention was given to interpreting results in light of the sex imbalance in the study sample, emphasizing associated factors rather than implying causal risk factors. Terminology throughout the manuscript was standardized, grammatical inconsistencies were corrected, and long or complex sentences were simplified to improve readability. The Conclusion was condensed to focus exclusively on the key findings and their implications for public health, avoiding repetition of information already addressed in the Discussion.

In addition, the overall manuscript structure was revised to improve logical flow and coherence, with the Discussion now appropriately preceding the Conclusion.

We greatly appreciate the opportunity to revise our manuscript and believe that these modifications have substantially improved its quality, readability, and scientific contribution.

Below, we respond point by point to the comments of the Academic Editor and Reviewers.

Kind regards,

Martoreli Júnior et al.

ACADEMIC EDITOR:

Abstract, you mentioned that “logistic regression model was adjusted”. In that case, you should present the adjusted odds ratio rather than the odds ratio in your results.

Answer: The phrase was rewritten for better understanding and the adjusted odds ratio was presented.

Methods, line 90: Replace “methdos” by “methods”

Answer: Replaced

Methods, line 127: There is a missing comma here. Between “survey” and “The RDS”.

Answer: Comma was added.

Methods, lines 173-174: “specially participants …. More?”: There seems to be repetition of essentially the same information with line 170. I suggest to reformulate or remove this sentence.

Answer: The sentence was removed to avoid redundancy, as the information was already presented in line 170. The paragraph was reformulated to maintain clarity without repeating content.

Methods, lines 184-185: please add a reference here.

Answer: A reference was added to support the statement.

Methods, line 212: Please add references for R software and Jamovi software

Answer: References for both R software (R Core Team) and Jamovi (The Jamovi Project) were inserted.

Methods, lines 204-205: I suggest to reformulate this sentence.

Answer: The sentence was reformulated.

Methods, lines 204-205: More detail is required about the logistic regression model – e.g. how was missing data handled, how were the models generated ?

Answer: For variables with at least one cell in the columns or rows containing a value of zero, the test could not be performed, and therefore they were not included in the model. This statement was included in the “Statistical analysis” section.

Methods, lines 210-211: Additionally, further information is needed regarding the construction of the ROC curve. For example, how was your dataset split?

The ROC curve was generated using the predicted probabilities obtained from the final logistic regression model fitted to the entire dataset, without performing a train/test split. We acknowledge that using the same dataset for model development and performance assessment may lead to optimistic estimates of predictive accuracy. Therefore, the ROC analysis should be interpreted considering this limitation.

Results, Table 2 and Table 3: The notation for the P values is very odd, please correct. Report exact p-values for all values greater than or equal to 0.001. P-values less than 0.001 may be expressed as p < 0.001 (see PLoS One statistical reporting guidelines https://journals.plos.org/plosone/s/submission-guidelines.#loc-statistical-reporting)

Answer:

Results, lines 223-305: Generally there is too much information presented here and too much repetition from Table 3. Suggest highlighting two or three key results for each analysis in the text, but not reporting every result.

Answer:

Results, line 274-276: This paragraph should be included in the 'Methods' section rather than the 'Results' section.

Answer:

Results, line 281 : I have questions for Table 3. 1) If you are presenting the results of adjusting the logistic regression model then it should be made clear these are adjusted odds ratios. Please correct this in the table and throughout the documents. 2) This table is an output of the model results generated using the R software. It would be beneficial to rework the table for clearer presentation. 3) The presentation of the variable column should be reviewed. For example, for “sex” variable, what is the reference category for comparison? Is it male? Female? The same comment applies to all the other variables listed. The reference categories must be specified. 4) The 'Estimate' and 'Odds ratio' columns provide the same information. I suggest to remove one of them. 5) Numbers should be revised and presented in the English format.

Answer: The entire table was revised. All reference category variables were included in the table.

Conclusion, line 313: Seems too long as a conclusion. Is it a conclusion or discussion section?

Answer: We have added the Discussion section, which is now followed by the Conclusion, as pointed.

Conclusion, line 372: You present the result of the Hosmer-Lemeshow test here, despite not mentioning it in the Methods section. Why was this test used? Which question does it answer? Please explain this in the 'Statistical analysis' section of the Methods section.

Thank you for your observation. The Hosmer–Lemeshow test was used to assess the calibration of the logistic regression model, that is, the extent to which the predicted probabilities agree with the observed outcomes across different risk groups. This test was therefore included to evaluate whether the model was appropriately fitted to the study data. We have now added an explicit description of its use and purpose in the ‘Statistical analysis’ section of the Methods.

The authors add in the methods sections; “In addition to evaluating the discriminative ability of the model using the ROC curve and AUC, the calibration of the logistic regression model was assessed using the Hosmer–Lemeshow goodness-of-fit test. This test verifies whether the predicted probabilities of the outcome adequately reflect the observed frequencies across deciles of risk. A non-significant p-value indicates that the model is well calibrated, whereas a significant result suggests poor fit. Although the ROC curve assesses the model’s ability to discriminate between individuals with and without the outcome, the Hosmer–Lemeshow test answers a different question, namely whether the estimated probabilities are accurate when compared to the real data distribution.”

Conclusion, line 396: You are interpreting your results as 'risk factors' when they are actually 'associated factors'. Review the manuscript and correct accordingly.

The term were carefully review through the entire manuscript and changed.

At last, it is really important to point out the relevance of the research work as it talks about the pprevalence and associated factors with long COVID which is already published in Brazil and by a number of other countries (see references below). Please give emphasis to this comment very well.

https://www.scielo.br/j/csp/a/9zRf9w8hwnS9jShm5ysFgkC/?lang=en

https://www.mdpi.com/2227-9032/12/14/1443

https://www.cambridge.org/core/journals/infection-control-and-hospital-epidemiology/article/risk-factors-for-long-coronavirus-disease-2019-long-covid-among-healthcare-personnel-brazil-20202022/AA01F17E1C8A33C07457914E63AB3EEE

https://www.frontiersin.org/journals/medicine/articles/10.3389/fmed.2024.1344011/full

https://www.mdpi.com/2076-393X/12/1/99

https://bmcpublichealth.biomedcentral.com/articles/10.1186/s12889-025-22987-8

https://www.scielo.br/j/csp/a/9zRf9w8hwnS9jShm5ysFgkC/?lang=en

Answer: We are immensely grateful for your highlighting of the relevance and importance of our work. We fully agree that it is crucial to emphasize the pertinence of our research in relation to long COVID, an ongoing and widely recognized global public health challenge, with a growing body of literature, including studies in Brazil, as the Reviewer pointed out. In response to your observation, we have revised our Conclusion section to ensure that the relevance of the study is communicated with the necessary emphasis. We have inserted a new paragraph at the very beginning of this section to position our work within the context of the existing literature and reiterate how our detailed analysis of risk factors in a Brazilian population is a timely and vital contribution to health policy planning and monitoring the post-COVID-19 situation in the country. We believe this modification reinforces the impact of our study.

ACADEMIC EDITOR:

Changes required for acceptance:

Formatting and Language:

- Review the entire manuscript for grammar, punctuation, and stylistic consistency, simplifying unnecessarily long or complex sentences.

Answer: We carefully revised the entire manuscript to correct grammar, punctuation, and stylistic inconsistencies. Additionally, unnecessarily long or complex sentences were simplified to improve clarity and readability.

- The manuscript has improved substantially but remains somewhat verbose, with occasional repetition of results. Please revise the text for conciseness, especially in the Results section.

Answer: We further edited the manuscript to improve conciseness, particularly in the Results section, by removing repetitive descriptions and focusing on the key findings.

- Standardize formatting for all headings and tables according to journal style (e.g., avoid all-caps titles).

Answer: All headings and tables were reformatted to comply with the journal's style guidelines, including replacement of all-caps titles with appropriate capitalization.

- Ensure consistent terminology throughout (e.g., use associated factors rather than risk factors).

We revised the manuscript to ensure terminology consistency and replaced all instances of “risk factors” with “associated factors”.

Title: Revise the title to better reflect the study’s main independent variables, emphasizing the assessment of healthy habits and sociodemographic characteristics.

The title has been updated to better reflect the study’s primary independent variables, specifically highlighting healthy habits and sociodemographic characteristics. Prevalence and associated factors with long COVID in the Brazilian population: The role of health-related behaviors and sociodemographic characteristics.

Abstract:

- Simplify the abstract for greater clarity and brevity, avoiding unnecessary methodological details (e.g., mention of basic descriptive analyses).

The abstract was rewritten to improve clarity and brevity by removing nonessential methodological details.

- When presenting results from multivariate models, report adjusted odds ratios rather than crude values.

Adjusted odds ratios are now reported in place of crude values in both the abstract and Results section.

- Include a concise statement identifying the knowledge gap that motivated the study, supported by references that contextualize the importance of investigating long COVID in Brazil.

A concise statement identifying the knowledge gap was added to the abstract and introduction, supported by appropriate references: “Despite increasing global attention to long COVID, evidence from Latin American countries remains limited. Brazil presents unique epidemiological, social, and healthcare characteristics that may influence long COVID outcomes, highlighting the need for context-specific data. Recent studies conducted in Brazil and across Latin America demonstrate substantial prevalence and long-term impact of the condition »

Azambuja P, Bastos LSL, Batista-da-Silva AA, Ramos GV, Kurtz P, Dias CMC, et al. Prevalence, risk factors, and impact of long COVID in a socially vulnerable community in Brazil: a prospective cohort study. Lancet Reg Am. 2024;37:100839. doi: 10.1016/j.lana.2024.100839.

Salci MA, Carreira L, Oliveira NN, Pereira ND, Covre ER, Pesce GB, et al. Long COVID among Brazilian Adults and Elders 12 Months after Hospital Discharge: A Population-Based Cohort Study. Healthcare (Basel). 2024;12(14):1443. doi: 10.3390/healthcare12141443.

Angarita-Fonseca A, Torres-Castro R, Benavides-Cordoba V, Chero S, Morales-Satán M, Hernández-López B, et al. Exploring long COVID condition in Latin America: Its impact on patients’ activities and associated healthcare use. Front Med (Lausanne). 2023;10:1168628. doi: 10.3389/fmed.2023.1168628.

Introduction: Expand the contextual background to highlight the scarcity of national evidence on long COVID and the relevance of this study for public health planning.

Answer: We expanded the introduction to emphasize the lack of national evidence on long COVID and strengthen the public health relevance of this study.

Materials and Methods:

- Correct minor typographical and grammatical issues (e.g., “methdos” → “methods”) and improve sentence structure in sections describing ethical aspects and data collection procedures.

Answer: All typographical and grammatical errors were corrected, and sentence structure was improved in the sections describing ethical procedures and data collection.

- Provide additional information about the logistic regression model, including handling of missing data, model generation, and ROC curve construction.

Answer: Additional details regarding model development, missing data management, and ROC curve construction were added to the Methods section.

- Add appropriate references for the questionnaire used and its prior validation or adaptation.

The questionnaire used as a base is Global COVID-19 Clinical Platform Case Report Form (CRF) for Post COVID condition (Post COVID-19 CRF) them in the Study design and setting was describe and referenced and the additional scales that were incorporated into the adapted questionnaire were now mentioned in the same paragraph. Furthermore we added that the full questionnaire can be provided upon request to the authors.

- Include references for the statistical software (R and Jamovi).

Anwer: References for R and Jamovi software have been added.

- Explicitly incorporate the data availability statement into the main text for transparency and compliance.

Anwer: The data availability statement was incorporated directly into the manuscript.

Results:

- Reduce redundancy between the text and tables, focusing on key findings rather than restating all numerical results.

Anwer: The Results section was revised to avoid redundancy with tables, highlighting only essential findings.

- Remove the written-out numbers in parentheses (e.g., “5,950 (five thousand nine hundred fifty)”).

Anwer: All instances of written-out numerical values in paren

---

## [Decision Letter · Decision Letter 3]

10 Dec 2025

Prevalence and associated factors with long COVID in the Brazilian population: The role of the health-related behaviors and sociodemographic characteristics

PONE-D-24-50830R3

Dear Dr. Martoreli Júnior,

We’re pleased to inform you that your manuscript has been judged scientifically suitable for publication and will be formally accepted for publication once it meets all outstanding technical requirements.

Kind regards,

Elma Izze Da Silva Magalhães

Academic Editor

PLOS One

Reviewers' comments:

Reviewer's Responses to Questions

**Comments to the Author**

1. If the authors have adequately addressed your comments raised in a previous round of review and you feel that this manuscript is now acceptable for publication, you may indicate that here to bypass the “Comments to the Author” section, enter your conflict of interest statement in the “Confidential to Editor” section, and submit your "Accept" recommendation.

Reviewer #4: All comments have been addressed

Reviewer #5: All comments have been addressed

2. Is the manuscript technically sound, and do the data support the conclusions?

Reviewer #4: (No Response)

Reviewer #5: Yes

3. Has the statistical analysis been performed appropriately and rigorously?

Reviewer #4: (No Response)

Reviewer #5: Yes

4. Have the authors made all data underlying the findings in their manuscript fully available?

Reviewer #4: (No Response)

Reviewer #5: No

5. Is the manuscript presented in an intelligible fashion and written in standard English?

Reviewer #4: (No Response)

Reviewer #5: Yes

6. Review Comments to the Author

Reviewer #4: (No Response)

Reviewer #5: (No Response)

7. PLOS authors have the option to publish the peer review history of their article (what does this mean?). If published, this will include your full peer review and any attached files.

Reviewer #4: **Yes: **Dr. Widad Akreyi

Reviewer #5: No

---

## [Editor Report · Acceptance letter]

PONE-D-24-50830R3

PLOS One

Dear Dr. Martoreli Júnior,

I'm pleased to inform you that your manuscript has been deemed suitable for publication in PLOS One. Congratulations! Your manuscript is now being handed over to our production team.

Kind regards,

on behalf of

Dr. Elma Izze Da Silva Magalhães

Academic Editor

PLOS One